# Enhanced Luminance of CdSe/ZnS Quantum Dots Light-Emitting Diodes Using ZnO-Oleic Acid/ZnO Quantum Dots Double Electron Transport Layer

**DOI:** 10.3390/nano12122038

**Published:** 2022-06-14

**Authors:** Da Young Lee, Hong Hee Kim, Ji-Hyun Noh, Keun-Yong Lim, Donghee Park, In-Hwan Lee, Won Kook Choi

**Affiliations:** 1Center for Optoelectronic Materials and Devices, Korea Institute of Science and Technology, Seoul 02792, Korea; dayoung5178@naver.com (D.Y.L.); hongheekim85@kist.re.kr (H.H.K.); luzhexian@hanyang.ac.kr (J.-H.N.); t14822@kist.re.kr (K.-Y.L.); 2Department of Materials Science and Engineering, Korea University, Seoul 02841, Korea; ihlee@korea.ac.kr; 3Department of Nanoscience and Engineering, KIST School, University of Science and Technology, Daejeon 34113, Korea

**Keywords:** QLEDs, electron transport layer, ZnO-OA QDs, double ETL, hole blocking

## Abstract

The widely used ZnO quantum dots (QDs) as an electron transport layer (ETL) in quantum dot light-emitting diodes (QLEDs) have one drawback. That the balancing of electrons and holes has not been effectively exploited due to the low hole blocking potential difference between the valence band (VB) (6.38 eV) of ZnO ETL and (6.3 eV) of CdSe/ZnS QDs. In this study, ZnO QDs chemically reacted with capping ligands of oleic acid (OA) to decrease the work function of 3.15 eV for ZnO QDs to 2.72~3.08 eV for the ZnO-OA QDs due to the charge transfer from ZnO to OA ligands and improve the efficiency for hole blocking as the VB was increased up to 7.22~7.23 eV. Compared to the QLEDs with a single ZnO QDs ETL, the ZnO-OA/ZnO QDs double ETLs optimize the energy level alignment between ZnO QDs and CdSe/ZnS QDs but also make the surface roughness of ZnO QDs smoother. The optimized glass/ITO/PEDOT:PSS/PVK//CdSe/ZnS//ZnO-OA/ZnO/Ag QLEDs enhances the maximum luminance by 5~9% and current efficiency by 16~35% over the QLEDs with a single ZnO QDs ETL, which can be explained in terms of trap-charge limited current (TCLC) and the Fowler-Nordheim (F-N) tunneling conduction mechanism.

## 1. Introduction

In various modern optoelectronic devices such as light-emitting diode (LEDs) [1,2,3] and photovoltaic cells [4], the II-VI ZnO oxide semiconductor has been widely investigated as an electrically superior ETL material due to a two order higher electron mobility of 10^−3^–10^−1^ cm^2^/Vs than other conventional TiO_2_ because the unoccupied Zn 4s orbitals with spherical symmetry consist of the bottom of the conduction band (CB) [5,6]. ZnO ETL-based QLEDs rapidly reached high performance with an external quantum efficiency (EQE) of 20% and lifetimes (>10^5^ h at 100 cd/m^2^) comparable to those of state-of-the-art OLEDs [7]. Besides ZnO QDs, a co-doped type of Zn (Ga, Mg)O QDs [8,9], as well as hybrid types of ZnO@TiO_2_ [10], core-shell QDs were adopted as ETL to improve the performance of CdSe/CdS-based QLEDs. The hybrid type of ZnO@SnO_2_ [4] was also used to improve the efficiency of inorganic perovskite solar cells. Double layer structured ETL of ZnO/PEIE (Polyethylenimine ethoxylated) [11] by lowering the conduction band maximum (CBM) of ZnO QDs ETL increases the luminance of inverted CdSe/ZnS QD-LEDs. Moreover, the double-layered ETL of ZnO/C_60_ could increase the luminance of perovskite LEDs (PeLEDs) by suppressing the exciton quenching caused by electron accumulation at the emissive layer (EML)/hole transport layer (HTL) interface, further increasing the exciton recombination inside the EML of CdSe/ZnS [12]. In the case of ETL materials, besides their excellent electrical properties, the electronic band alignment with EML and solution processability are crucial factors in determining the device efficiency. Regarding band alignment, however, one ZnO QDs ETL drawback is that charge balancing of electrons and holes is not taking place effectively due to the low hole blocking efficiency caused by the small energy difference between the VB (6.38 eV) of ZnO ETL and that (6.3 eV) of CdSe/ZnS. Surface modifications of ZnO with the organic ligands such as oleic acid (OA), oleylamine (OLA), and stearic acid (SA) have been utilized to increase the spatial distribution in an anti-solvent of perovskite EML to enhance the electrical contact with ZnO ETL [13] as well as for controlling of the defects existing at the surfaces to change the optical properties [14,15] (Appendix A). In this study, ZnO QDs are synthesized by solution precipitation process using Zn acetate (Zn(CH_2_COOH)_2_) and tetramethylammonium hydroxide (TMAH) as a reducing agent and DMSO as a precursor at 70 °C. Afterward, the ZnO-OA QDs were synthesized by the chemical reaction of the oleic acid ligands (0–1.5 mL) with ZnO QDs. Besides simple conventional segregation of ZnO QDs through surface capping of OA ligand, we investigated how the OA ligand interacts with the ZnO surface defects and how this influences the work function for the electron injection, the VB for the hole blocking efficiency, band gap values for photoluminescence (PL), and ETL properties including the solvent orthogonality and the spatial distribution between EML and ZnO-OA QDs. Glass/ITO/PEDOT:PSS/PVK//CdSe/ZnS//ZnO-OA/ZnO/Ag QLEDs devices are fabricated using spin-coating and thermal evaporation. Enhanced luminance of CdSe/ZnS QD-LEDs with a double ETL of ZnO-OA/ZnO QDs compared to those with a single ZnO, or ZnO-OA QDs ETL, is well elucidated in terms of space-charge-limited current conduction (SCLC), trap-charge-limited-current (TCLC), and Fowler-Nordheim (F-N) tunneling conduction mechanism. Additionally, comparing an electron-only device (EOD) and a hole-only device (HOD) verifies the enhanced electron transport capability and the improvement of the charge balance in QLEDs with double ETLs compared to those with a single ETL.

## 2. Materials and Methods

### 2.1. Synthesis of ZnO Quantum Dots

ZnO QDs were prepared via a low-cost solution precipitation process using Zn acetate (Zn(CH_2_COOH)_2_) (99.99%) and tetramethylammonium hydroxide (TMAH, ≥97%) [16,17]. Here, 5 μM TMAH in 10 mL of pure ethanol (solution A) as a reducing agent and 3 μM Zn acetate in 30 mL of dimethyl sulfoxide (DMSO, ≥99.7%) (solution B) as a precursor were dissolved completely at room temperature for 30 min. Solution B was added slowly dropwise into solution A for 1 h to produce ZnO QDs at 70 °C. The ZnO QDs were washed with excess acetone, and the resultant ZnO QDs were collected via centrifugation in the solution at 12,000 rpm for 10 min, after which they were ultrasonically redispersed in pure 15 mL ethanol for use.

### 2.2. Synthesis of ZnO-Oleic Acid (OA) QDs

The synthesized ZnO QDs were dispersed in 100 mL of ethanol at room temperature. Then, a solution was added of Oleic Acid (0.5, 1.0, and 1.5 mL) in 30 mL of ethanol for 1 h to produce ZnO-OA core-shell QDs at 120 °C with a magnetic stirrer. Finally, the ZnO-OA core-shell QDs were washed with excess acetone and ethanol. The resultant ZnO-OA QDs were collected via centrifugation in the solution at 3000 rpm for 10 min, after which they were dispersed in 2-propanol (~5 mg/mL).

### 2.3. QLEDs and Electron (Hole)-Only Device Fabrication

The multilayer-structured EL device of QLEDs was fabricated into the multilayered structure using a spin-casting method and a vacuum evaporation method. Before the device fabrication step, the patterned ITO substrates were cleaned by sonication sequentially in acetone, methanol, and isopropyl alcohol. The ITO substrates were then rinsed with D.I. water and treated with O_2_ plasma. The final step involved PEDOT:PSS (Clevious^TM^, PH-1000, Heraeus, Hanau, Germany) polymer in isopropyl alcohol (2.39 wt.%) solution, which was mixed for 15 min and spin-coated onto the ITO/glass as a hole injection layer (HIL). This was then dried at 110 °C on a hot plate for 30 min. In a nitrogen glovebox, PVK (M_n_ 25,000–50,000, Sigma-Aldrich, St. Louis, MO, USA) was spin-coated on top of the PEDOT:PSS layer at 5000 rpm for 40 s and annealed at 130 °C for 30 min, followed by spin-coating of red QDs (CdSe/ZnS) in a hexane solution (20 mg/mL) on the top and subsequent annealing at 90 °C for 30 min as an EML of the EL device. And then, ETL of ZnO QDs, ZnO-OA QDs, or double ZnO-OA/ZnO QDs was deposited on the EML using spin-coating at 3000−5000 rpm for 40 s and baked at 130 °C for 30 min. Finally, an Ag cathode film (120 nm) was deposited by thermal evaporation through a patterned shadow mask. The schematic diagrams of the cross-sectional view of CdSe/ZnS core-shell-based red QLEDs with a single and double ETL are presented in Figure 1. Glass lid encapsulation was executed using a UV-curable sealant (XNR5570-Ba, Nagase Chemtex, Osaka, Japan) for I-V-L measurement. Furthermore, the EOD of ITO/PEIE//CdSe/ZnS//ZnO (or ZnO-OA/ZnO)/Ag and the HOD of ITO/PEDOT:PSS/PVK//CdSe/ZnS//MoO_3_/Ag were fabricated respectively (Figure 2) to identify the conduction mechanism responsible for the enhanced current efficacy.

### 2.4. Characterization

The crystalline structure of ZnO and ZnO-OA QDs was characterized by X-ray diffraction and high-resolution transmission electron microscopy (HR-TEM). The X-ray diffraction (XRD) patterns were recorded using an X-ray diffractometer (Rigaku ATX-G) with Cu *Kα* radiation of wavelength λ = 1.5406 Å in the scan range 2θ = 20° to 80°. TEM samples were prepared by dispersing QDs dry powders in deionized water to form a homogeneous suspension. HR-TEM and diffraction pattern analyses were carried out using an FEI Super-Titan (TM80-300) STEM/TEM operating at 300 keV. The size distribution of ZnO and ZnO-OA QDs was analyzed using Gatan Microscopy Suite Software (DigitalMicrograph, Ver.2.11.1404.0). Fourier transform-infrared (FT-IR) spectroscopy was carried out using a Perkin Elmer Spotlight 400N FT-NIR Imaging System. UV-visible absorption data were recorded by a PerkinElmer Lambda 18 UV-vis spectrometer with QS grade quartz cuvettes. Spectral-dependent PL (SDPL) and photoluminescence excitation (PLE) data were recorded by a Hitachi F-7000 fluorescence system with QS-grade quartz cuvettes. The work function (φ) of ZnO and ZnO-OA QDs was measured by ultraviolet photoemission spectroscopy (UPS: Nexsa, ThermoFisher Scientific (Hong Kong, China)) using He I (hν = 21.22 eV) as a light source a hemispherical energy analyzer with the energy resolution of 100 meV and pass energy of 5.0 eV, and defined as the energy difference between the vacuum level (E_vac_ = 0 eV) and the Fermi energy (E_F_). The I-V-L characteristics of the EL device were analyzed using a SpectraScan PR-670 spectroradiometer and a Keithley 2400 source-measure unit.

## 3. Results and Discussions

### 3.1. Synthesis of ZnO Quantum Dots

Figure 3a represents the XRD patterns of the synthesized ZnO and ZnO-OA QDs. XRD peaks observed at 2θ values of 31.66°, 34.52°, 36.2°, 47.7°, 56.64°, 62.96°, 66.62°, 67.9°, and 69.02° can be indexed to different planes (100), (200), (101), (102), (110), (103), (200), (112), and (201) of crystalline ZnO with hexagonal wurtzite structure (JSPDF File No. 79-2205). The ZnO-OA (0.5 mL) QDs show a decrease in the degree of crystallinity and many sharp diffraction peaks around 2θ = 20~40° due to the microcrystalline-like structure of OA ligands. Figure 3b is the images of TEM observed to confirm the size of the synthesized ZnO and ZnO-OA QDs. The average size of the ZnO QDs is about D_a_ = 7.2 nm, and that of the ZnO-OA (0.5 mL) is approximately D_a_ = 7.3 nm. As the amount of the OA ligand is further increased, the average size of ZnO-OA (1.0 mL) QDs is slightly reduced to about D_a_ = 6.9 nm, and the crystallinity is also lowered. However, if the amount of OA is added up to 1.5 mL, the average size is quite reduced to about D_a_ = 5.4 nm, and the crystalline phases of ZnO QDs almost disappear. While forming ZnO-OA quantum dots via hydrothermal process at 120 °C, Zn^2+^ from the ZnO QDs surface dissociates to form Zn-OA and then dissolved in solvent or redeposited on the surface. As the amount of OA ligand attached to the surface of ZnO QDs increases, the number of Zn-OA dissolved in the solvent becomes larger than the number redeposited on the ZnO surface, so the size of ZnO QDs decreases, and ZnO QDs also decrease.

### 3.2. Surface Chemistry

From the FT-IR spectrum of ZnO QDs (Figure 4), both hydroxyl (-OH) and acetoxy (-O-C=O-CH_3_) groups on the surface of ZnO QDs are identified by the observation of absorption peaks at 3650–3200 cm^−1^, 1560 cm^−1^ and 1402 cm^−1^, which correspond to the stretching vibrations of O-H group, the asymmetric and symmetric stretching vibrations of COO, respectively. The absorption peaks at 1340 cm^−1^ and 1488 cm^−1^ correspond to the C–N bonding of TMAH, indicating the remains of the base product TMAH within ZnO QDs. These peaks almost disappear after the OA ligand reacts with the ZnO QDs in ethanol through heating at 120 °C. The ZnO-OA (0.5 mL) core-shell QDs show FR-IR absorption peaks at 2926 cm^−1^ (sp^2^ C-H stretch, -CH_3_), 2854 cm^−1^ (sp^3^ C-H stretch, -CH_2_), 1540 cm^−1^ (COO-), 1464 cm^−1^ (CH_3_ bend), 1406 cm^−1^ (CH_2_ bend), and 745 cm^−1^ (CH_2_ rock, bending) which are all characteristic of the OA ligand (2CH_3_(CH_2_)_15_COOH). This FT-IR spectrum is very similar to those previously reported [18,19]. The vibration band observed at 450~460 cm^−1^ corresponds to the stretching vibration of the zincite structure (Zn-O) [20]. The FT-IR spectrum of the ZnO-OA (1.0 mL) QDs is almost the same as that of the ZnO-OA (0.5 mL) QDs except for the slight increase in the intensity.

Figure 5 presents the survey and core-level XPS spectra of the ZnO and ZnO-OA (0.5 mL, 1.0 mL) QDs. From the survey spectra (Figure 5a), the intensity of C1s in the ZnO-OA QDs largely increases, but that of Zn2p and O1s decreases compared to ZnO QDs. C1s XPS peak (Figure 5b) for ZnO QDs can be fitted into three sub-peaks at binding energy (BE) = 284.67 eV, 286.28 eV, and 288.21 eV, which correspond to C-CH_3_ and N-CH_3_/OH-CH_3_ of TMAH, and the acetoxy group of O-C(-CH_3_)=O, respectively. As the result of the chemical bonding of the OA ligand, C1s BE of ZnO-OA QDs shows a red shift towards a little lower BE side due to the increase of sp^2^ and sp^3^ C-H bond. Figure 5c shows the O1s core-level XPS peaks of ZnO and ZnO-OA QDs. The intensity of the sub-peak related to Zn-O around 529.42–529.91 eV is gradually decreased, but that related to Zn acetate (Zn(CH_3_COO)) or Zn-OA (Zn-OOC(C_17_H_33_)) around 532.14–532.56 eV progressively increases. O1s peak for ZnO QDs is resolved into lattice O (Zn-O) at BE = 529.42 eV, oxygen vacancy (V_O_) at BE = 530.19 eV, hydroxyl Zn-OH at BE = 531.22 eV, and Zn acetate at BE = 532.14 eV. (Appendix A) The relative chemical bond ratio is approximately 49.7%:20.5%:22.0%:7.8% (Appendix A). In the case of ZnO-OA (0.5 mL), the relative chemical bond ratio becomes Zn-O:V_O_:Zn-OH:Zn-OA = 9.9%:20.6%:54.2%:15.3%. But, in the case of ZnO-OA (1.0 mL), the chemical bond of Zn-OH disappears, and only Zn-O, V_O_, and Zn-OA are included, and the chemical bond ratio is estimated as 2.9%:20.6%:76.5%. Figure 5d shows the Zn2p core-level XPS peak of ZnO and ZnO-OA QDs. The spin-orbit splitting of ∆_SO_ = 23.09 eV between Zn2p^1/2^ (1044.02 eV) and Zn2p^3/2^ (1020.93 eV) for ZnO QDs is the same as that ∆_SO_ = 23.09 eV of ZnO-OA QDs. However, Zn2p of ZnO QDs peak is blue-shifted as ∆E = 0.4 eV and 1.0 eV in ZnO-OA (0.5 mL, 1.0 mL) QDs respectively, which means that electrons are transferred from inner ZnO (donor) core to outer OA (acceptor) shell.

### 3.3. Optical Properties

Figure 6 presents the Tauc plot ((α*h*ν)^2^ vs. *h*ν) obtained by using Equation (1) from the UV-Vis absorption spectrum of ZnO and ZnO-OA QDs.
(1)$$\alpha h\nu =A{\left(h\nu -E_{g}\right)}^{\frac{1}{2}}$$
where α is an absorption coefficient, *h* is Plank’s constant, ν is photon frequency, *A* is the constant relating the effective mass associated with valence and conduction bands, and *E*_g_ is the optical band gap. From the Tauc plot, *E*_g_ is determined by extrapolating the linear part until it intersects with the *h*ν axis. The *E*_g_ of ZnO and ZnO-OA (0.5 mL, 1 mL) QDs is estimated as 3.3 eV, 3.4 eV, and 3.81 eV, respectively.

Figure 7a presents SDPL for ZnO QDs in the excitation wavelength (λ_ex_) of 300~430 nm. A typical yellow luminescence (YL) is observed at 562 nm (2.20 eV) and shows the full width at a half maximum (FWHM) of about 96 nm and the maximum intensity at λ_ex_ = 380 nm (3.26 eV) corresponding to E_g_ = 3.3 eV of ZnO QDs. YL is still observed until λ_ex_ = 400 nm (3.10 eV), but almost disappears at λ_ex_ > 400 nm. But in the range of λ_ex_ = 410~430 nm, only the sharp blue luminescence (BL) with a relatively narrow FWHM is newly observed at about 460 nm. These results indicate that YL and BL depend directly on the excitation wavelength and have different luminescence centers. In the case of YL, excitation energy (E_ex_) should be larger than the threshold energy (E_th_) of 3.10 eV (i.e., the threshold excitation wavelength, λ_th_ = 400 nm), which energy range corresponds to the CB and shallow donor levels like Zn interstitial (Zn_i_) located within 0.3 eV below CB. However, BL can only be observed at excitation energy less than E_ex_ < 3.02 eV (410 nm). From the photoluminescence excitation (PLE) spectra for ZnO QDs, it is revealed that YL is derived by excitation only from an energy level of 375 nm (3.30 eV) corresponding to the CB, and BL is derived by two energy levels of both 375 nm and 430 nm (2.88 eV). Consequently, it can be said that YL correlates with the electronic transition from the shallow donor levels of Zn_i_ to deep levels at about 0.9 eV from the VB, which is known as the intrinsic defect of doubly ionized Zn vacancy (V_Zn_^2−^) or vacancy (V_O_). In YL, electrons can be excited to Zn_i_ directly or through a non-radiative transition from the CB. BL of 460 nm (2.69 eV) originates from the electronic transition from ex-Zn_i_ to V_Zn_^−^. Similarly, electrons can be excited to ex-Zn_i_ directly or through a non-radiative transition from the CB. These results are consistent with the previous PL for the ZnO QDs [16,17]. As can be seen in Figure 7b,c, the λ_ex_ showing the maximum PL intensity of ZnO-OA (0.5 mL, 1 mL) QDs is blue-shifted to 360 nm (3.44 eV) and 330 nm (3.75 eV) as shown in Commission Internationale de l’Eclairage (CIE) 1931 (Figure 7d), which is coincident with E_g_ = 3.4 eV and 3.81 eV, respectively.

### 3.4. Electronic Energy Level Structure

Figure 8 shows UPS spectra of ZnO and ZnO-OA (0.5 mL, 1.0 mL) QDs. From the UPS spectra, the work function (φ) of ZnO and ZnO-OA QDs are estimated as the values of 3.15, 3.08, and 2.72 eV, respectively (Figure 8a,c,e), by extrapolating the cutoff curve of the kinetic energy of secondary electrons using the relation φ = hv − (E_cutoff_ − E_F_), where hv = 21.22 eV (corresponding to the He I UV resonance line). The energy level of the VB is also acquired using the relationship VB = 21.22 − (E_cutoff_ − E_onset_), where E_onset_ is the energy onset in the VB region. Thus, the energy of the VB is calculated as 3.25 eV for ZnO QDs, and 4.15 eV and 4.5 eV for ZnO-OA (0.5 mL, 1 mL) QDs. Accordingly, the energy level of the VB can be defined as −6.4 eV for ZnO QDs and −7.23 eV for ZnO-OA (0.5 mL), and −7.22 eV for ZnO-OA (1 mL) QDs from E_vac_ (0 eV) respectively.

Considering E_g_ obtained from Tauc’s plot (Figure 6), the energy level of the CB can be assumed as −3.1 eV for ZnO QDs, and −3.83 eV for ZnO-OA (0.5 mL) and −3.41 eV for ZnO-OA (1 mL) QDs from E_vac_ (0 eV) respectively. From these results, an Anderson electronic energy diagram can be plotted in Figure 9. The lowering of φ = 3.15 eV in the ZnO QD to φ = 3.08 eV and 2.7 eV for ZnO-OA QDs can be reasonably interpreted by the formation of an induced surface dipole in the OA ligand through electron transfer from the inner ZnO QDs core to the outer OA shell, which is often found in previous reports [21,22] including the example of ZnO/PEIE [11].

### 3.5. Improved EL of CdSe/ZnS QLEDs by Optimization of ETL

#### 3.5.1. QLEDs with a Single ETL

EL performances of QD-LEDs with ZnO QDs ETL and ZnO-OA QDs ETL of different amounts of oleic acid (0.5 mL, 1.0 mL) at different thicknesses (3000~5000 rpm) are plotted in Figure 10. As shown in Table 1, the CdSe/ZnS QLEDs with a single ZnO QDs ETL (4000 rpm) shows the turn-on voltage at 3.5 V, the maximum luminance of 25,570 cd/m^2^ at 8 V, and the current efficiency of 6.32 cd/A respectively. On the other hand, all the QLEDs with a single ZnO-OA (0.5 mL) and ZnO-OA (1.0 mL) QDs ETL structure show quite inefficient EL performance compared to the QLEDs with ZnO QDs ETL only. Among them, turn-on voltage increases up to 8 V for ZnO-OA (0.5 mL) (5000 rpm) and 8.5 V for ZnO-OA (1.0 mL) (3000 rpm), and the maximum brightness largely decreases to the values of 814 cd/m^2^ and 691 cd/m^2^ at 10 V respectively. The current efficiency is also considerably reduced to 0.68 cd/A at 9.5 V and 0.53 cd/A at 9 V. As a result, the higher the amount of the OA ligand is added, the lower the luminance and current efficiency become irrespective of the thickness of the ZnO-OA QDs ETL. In the Schottky junction of a metal-semiconductor interface, the barrier height *φ_B_* for electrons can be easily calculated as the difference between the metal work function (*φ_M_*) and the electron affinity (χ) of the semiconductor as the Equation (2),
(2)$$\phi_{B}=\phi_{M}-\chi$$

When the junction of Ag/ETL is contacted, the Fermi level should be aligned, and then the built-in potential, *V_bi_*, is the difference between the Fermi energy of the metal and that of the semiconductor as expressed in the Equation (3),
(3)$$V_{bi}=\phi_{M}-\chi -\left(E_{CB}-E_{F}\right)$$

Figure 11 represents the electronic band diagram at the interface between ZnO QDs, ZnO-OA QDs (0.5 mL, 1.0 mL), and the Ag cathode. Calculated values of φ_B_’s are 1.2 eV, 0.48 eV, and 0.9 V for Ag/ZnO QDs, Ag/ZnO-OA (0.5 mL) QDs, and Ag/ZnO-OA (1.0 mL) QDs respectively. Similarly, the values of *V_bi_*’s are also calculated as 1.15 eV for ZnO QDs, 1.22 eV for ZnO-OA (0.5 mL) QDs, and 1.6 eV for ZnO-OA (1.0 mL) QDs respectively. Such an increase of *V_bi_* as much as 0.07 eV and 0.45 eV will further prevent electrons from injecting into the CdSe/ZnS QDs. As the barrier height of ZnO-OA (0.5 mL, 1.0 mL) QDs for hole injection increases by ~0.8 eV, the probability of recombining holes with electrons in the ETL layer is much lessened.

Accordingly, the large decrease in luminance of QLEDs with only ZnO-OA QDs ETL may be closely related to the reduction in the electrical conductivity of the ZnO-OA QDs ETL largely determined by the amounts of long-chain capping ligand OA and the decrease in the injection amounts of electrons. Judging from the above results, we propose a double ETL structure consisting of the ZnO-OA QDs with high hole blocking ability and simultaneous inhibition of electron injection and the ZnO QDs with high electron conductivity to investigate their characteristics in terms of electroluminescence and conduction mechanism.

#### 3.5.2. Interface Morphology

To examine the effect of ETL insertion on the interface morphology between CdSe/ZnS and ZnO QDs ETL, CdSe/ZnS QDs, and ETLs of ZnO-OA (0.5 mL, 1 mL)/ZnO QDs are successively spin-coated on glass/ITO substrate. Figure 12a,b show a cross-sectional HR-TEM images of the multilayered QLEDs devices of ITO/PEDOT:PSS/PVK//CdSe/ZnS//ZnO-OA QDs (0.5 mL, 1 mL)/ZnO QDs/Ag. The thickness of the PEDOT:PSS, PVK, CdSe/ZnS red QDs, and ZnO-OA (0.5 mL)/ZnO QDs, and Ag are estimated to be 18 nm, 25 nm, 9.2 nm, 23.65 nm, and 130 nm, respectively. Figure 12c presents EDS line mapping and line scan profile. From the EDS images of Cd L_α1_ and Zn K_α1_, it is revealed that a discrete interface is successfully formed at the boundary between CdSe/ZnS red QDs and ZnO-OA/ZnO double ETLs. As shown in Figure 12d–g, the average surface roughness (R_a_) of ITO//CdSe/ZnS QDs by scanning area over 5 μm × 5 μm is as smooth as 1.68 nm and largely increases to R_a_ = 16.3 nm for the ITO//CdSe/ZnS QDs/ZnO QDs, which might be caused by somewhat agglomeration of ZnO QDs. However, the ITO//CdSe/ZnS//ZnO-OA (0.5 mL)/ZnO QDs with double ETLs show a reduced surface roughness of R_a_ = 7.94 nm by almost 50%. But instead, when ZnO-OA (1.0 mL) QDs are inserted, R_a_ for the ITO//CdSe/ZnS//ZnO-OA (1.0 mL)/ZnO QDs returns to a slightly larger value of 14.77 nm. These results imply that the surface roughness of the ZnO QDs ETL directly coated on CdSe/ZnS QDs EML can be considerably reduced by the additional insertion of ZnO-OA QDs with a hydrophobic capping layer. Thus a low leakage current and a more successful charge carrier injection are expected, but it is closely dependent on the amount of the OA ligand.

#### 3.5.3. EL with a Double ETLs

To optimize the thickness of the ZnO-OA layer in the double ETL structure, several QLEDs are fabricated at the variously combined thickness of the ZnO QDs and ZnO-OA QDs layers and carefully evaluated. Among QLEDs (Figure 13a–c) having a single ZnO QDs ETL coated at the thickness of 3000–5000 rpm, the QLEDs with the thickness of 4000 rpm show a turn-on voltage of 4.5 V, the maximum luminance of 25,570 cd/m^2^ at 8 V, and current efficiency of 6.32 cd/A at 6 V. As summarized in Table 1, the QLEDs (Figure 13d–f) with double ZnO-OA (0.5 mL) (3000~5000 rpm)/ZnO QDs ETL show the lower turn-on voltage of 3.5 V than that of a single ZnO QDs ETL, the luminance of 23,760–27,890 cd/m^2^ and current efficiency (CE) of 5.08~8.57 cd/A.

Similarly, the QLEDs (Figure 13g–i) with double ZnO-OA (1.0 mL) (3000~5000 rpm)/ZnO QDs ETL also show a turn-on voltage of 3.5–4.5 V, luminance of 25,940~26,980 cd/m^2^ and current efficiency of 5.84–7.48 cd/A. In the QLEDs with double ETLs, the thinner the ZnO-OA layer becomes, the better the luminance and current efficiency of the device is attained. Both ZnO-OA (0.5 mL)/ZnO QDs and ZnO-OA (1.0 mL)/ZnO QDs show the same tendency.

Now the optimized QLEDs showing the best performance are designated as device 1 (ITO/PEDOT:PSS/PVK//CdSe/ZnS//ZnO QDs (4000 rpm)/Ag), device 2 (ITO/PEDOT:PSS/PVK//CdSe/ZnS//ZnO-OA (0.5 mL) QDs (5000 rpm)/ZnO QDs (4000 rpm)/Ag), and device 3 (ITO/PEDOT:PSS/PVK//CdSe/ZnS//ZnO-OA (1.0 mL) QDs (5000 rpm)/ZnO QDs (4000 rpm)/Ag) and are redrawn as like Figure 14. Among these devices, device 2 is considered the best EL performance QLEDs, demonstrating the highest luminance of 27,890 cd/m^2^ at 7.5 V and the maximum current efficiency of 8.57 cd/A at 6 V, which values correspond to an increment of 9% in luminance and 35.6% in current efficiency compared to device 1 respectively. After that, device 3 shows the luminance of 26,923 cd/m^2^ at 8 V and a current efficiency of 7.37 cd/A at 7 V, corresponding to an increment of 5% in luminance and 16% in current efficiency compared to device 1, respectively. Consequently, the QLEDs with ZnO-OA/ZnO double ETL structure significantly enhance device performance more than one with a single ZnO QDs ETL structure.

These results may be related to several factors. The first factor lies in the increment of potential barrier up to 7.22–7.33 eV between the VB of CdSe/ZnS QDs and that of ZnO-OA QDs, which will be superior to hole blocking compared to ZnO QDs. As the second factor, the insertion of ZnO-OA QDs not only facilitates electron injection between ZnO QDs and the CdSe/ZnS QDs EML, but also makes the surface roughness smoother, leading to low leakage current and high efficiency of injecting charge carrier. Figure 15 illustrates the schematic electron energy level of these double ETL structures of ZnO/ZnO-OA QDs with Ag cathode. The interface between ZnO QDs and ZnO-OA QDs in double ETL can be assumed as the junction of semiconductors with different electron affinity (χ). In the double ETLs structure, the *V_bi_* of 1.15 eV and 0.07 eV at Ag/ZnO and ZnO/Zn-OA (0.5 mL) interface and the *V_bi_* of 1.15 eV and 0.45 eV at Ag/ZnO and ZnO/Zn-OA (1.0 mL) interface are formed respectively. To find out exactly how the ZnO-OA/ZnO double ETLs act on both charge balancing and transport to further enhance QLEDs performances, an electron-only device (EOD) and a hole-only device (HOD) are fabricated and analyzed.

#### 3.5.4. EOD and HOD

A single ZnO QDs ETL and double ZnO-OA (0.5 mL (5000 rpm), 1.0 mL (5000 rpm)) ETLs are adopted for EODs, as described in Figure 3. They are designated as EOD-S with a single ETL and EOD-D1 with double ETL of ZnO-OA (0.5 mL)/ZnO QDs, and EOD-D2 with double ETL of ZnO-OA (0.5 mL)/ZnO QDs, respectively. The J-V characteristic curves of all the EODs are plotted in Figure 16. Overall, the electron current density for the EOD-S is higher than that for the EOD-D1 and EOD-D2 over the applied voltage below 1.3 V and 3 V, respectively. After those voltage ranges, the EOD-D1 and the EOD-D2 show almost the same analogous characteristic curves with the EOD-S until the voltage of 10 V. The EOD-D1 shows the current density value of one order less than those of the EOD-S and EOD-D2 and is closer to that of the HOD. This indicates that the EOD-D1 is capable of higher balancing charge carrier transport than the EOD-S and EOD-D2, and thus shows the more improved luminance and current efficiency. From the results, compared to the EOD-S, it is shown that the electron injection into emissive CdSe-ZnS QDs can be effectively modulated by the combination of ZnO-OA QDs with ZnO QDs as with the EOD-D1 and EOD-D2.

#### 3.5.5. Conduction Mechanism

To thoroughly examine the carrier transport phenomenon in QLEDs with a single ETL and double ETLs, we attempt to interpret the J-V characteristic curve in terms of three different conduction mechanisms [23,24]: thermionic emission (TE), space-charge limited current (SCLC) and/or trap-charge limited current (TCLC), and Fowler-Nordheim (F-N) tunneling. These three processes can be expressed in terms of Equations (4)–(6).
(4)$$J\;\propto A\;*\;T^{2}exp\left[-\frac{q\phi_{0}}{kT}+q{\left(\frac{q^{3}V}{4\pi \varepsilon }\right)}^{\frac{1}{2}}\right],\;\;\;$$
thermionic emission
(5)$$J\;\propto V^{m},\;\;\;$$
space charge limited current
(6)$$J\;\propto V^{2}\exp\left(\frac{-\kappa }{V}\right),\;\;\;$$

Fowler-Nordheim tunneling.

Figure 17a–d represent J-V, ln (J)-ln (V), and ln (J/V^2^)-1/V characteristic curves of the CdSe/ZnS QLEDs with a single ETL and double ETLs. The QLEDs with a single ETL were assigned as QLEDs-S1 (ZnO QDs ETL), QLEDs-S2 (ZnO-OA (0.5 mL) QDs ETL), and QLEDs-S3 (ZnO-OA (1.0 mL) QDs ETL), respectively. Similarly, QLEDs with double ETLs are also designated as QLEDs-D1 (ZnO-OA(0.5 mL)/ZnO QDs) and QLEDs-D2 (ZnO-OA (1.0 mL)/ZnO QDs), respectively. In Figure 17b, as in the EOD-S, the ln (J)-ln (V) for the QLEDs-S1 is divided into the SCLC region showing m_2_ = 1.9 at a bias voltage below 2.5 V, and the TCLC region in the voltage of 2.5~8 V consisting of two parts; one has the smaller values of m_3S_ = 3.71 in the voltage range of 2.5~5 V and the other has the larger values of m_3L_ = 10.96 in the voltage ranges of 5~8 V. After combining ZnO-OA (0.5 mL) QDs, the QLEDs-D1 shows a smaller J than QLEDs-S1 up to 2.5 V, but above that it shows almost the same value with QLEDs-S1. The slopes are fitted with m_2_ = 1.89, m_3S_ = 3.77, and m_3L_ = 13.8. On the other hand, the QLEDs-D2 shows a smaller J than QLEDs-S1 in the whole voltage range, and the slopes are fitted with m_2_ = 1.65, m_3S_ = 2.96, and m_3L_ = 14.8. The voltage ranges of 2.5~5 V showing the smaller slopes of m_3S_ in the TCLC region for QLEDs-S1, -D1, and -D2 are nearly identical to 3~5 V, showing the TCLC region for the EOD. In Figure 17c, the ln (J)-ln (V) curves of the QLEDs-S2 and -S3 show the analogous curves with the OLEDs-S1. The slopes of the power law fitted with the values of m_2_ = 1.91, 1.56, and 1.54 for the QLEDs-S1, -S2, and -S3 in the voltage ranges of 1~2.5 V are close to the ideal factor of m = 2 corresponding to the carrier transport by SCLC. On the other hand, the most striking feature is that the QLEDs-S2 and QLEDs-S3 without containing ZnO QDs show a negative slope of m_3N_ = −0.9 and m_3N_ = −0.97 in the voltage ranges of 3~7.5 V and after that voltage show again the positive slopes of m_3L_ = 15.3 and 17.2 respectively. For clear interpretation, the ln (J/V^2^)-1/V curves related to F-N tunneling for all the QLEDs are plotted in Figure 17d. The QLEDs-S1 and QLEDs-D3 show a positive linear slope in two voltage regions: one region in the voltage of 2.5~5 V shows the relatively small value of *k_S_* = 2 for QLEDs-D3 and *k_S_* = 2.6 for QLEDs-S1. The other region in the voltage ranges of 5.5~8 V shows a high value of *k_L_* = 27.52 for QLEDs-S1 and *k_L_* = 37 for QLEDs-D3, respectively. The slopes for QLEDs-D2 are well fitted with *k_S_* = 3 and *k_L_* = 22, and the voltage at which the low slope (*k_S_*) starts to shift toward a slightly larger value of 3.5 V from 2.5 V. It is noteworthy that the QLEDs-S3 shows the negative slope in the voltage regions of 2.5~7 V, in more detail *k_S1_* = −3.6 for 3~4.5 V and *k_S2_* = −8.6 for 4.5~7 V, respectively, and the positive slope of *k_S_* = 44.74 in the voltage of 7~10 V. In case of the QLEDs-S2, it is not easy to accurately determine the *k_S_* of the negative slope due to several kinks in the voltage ranges of 2.5~7 V, but the positive slope is fitted with *k_L_* = 44.74 in the voltage ranges of 7~10 V. From the ln (J/V^2^)-1/V curves, it can be claimed that the abrupt increase of J for the QLEDs-S1~S3 in the voltage regions of 5.5 V~8 V is closely related to F-N tunneling, which regions coincide with those showing m_3L_ in Figure 17b. In addition, the negative slope of m_3N_ = −0.9 and −0.97 for the QLEDs-S2 and -S3 in the voltage of 3~7.5 V is reasonably explained by the suppression of F-N tunneling.

At the Schottky junction, the width of the depletion layer (*W_D_*) [25] can be extracted as the Equation (7),
(7)$$W_{D}=\sqrt{\frac{2\varepsilon_{ZnO}V_{bi}}{eN_{d}}}$$
where *V_bi_* is the built-in potential at the boundary, *e* is the electronic charge, *ε_ZnO_* (=8.5) is the static dielectric constant of ZnO, and *N_d_* is the donor density. If it is assumed that *N_d_* of ZnO QDs is the same with ZnO-OA (0.5 mL, 1.0 mL) QDs and *ε_ZnO_* for ZnO and ZnO-OA QDs is 8.5, *W_D_* is simply proportional to (*V_bi_*)^1/2^. SCLC originates from the space charge and so depends on the magnitude of the depletion region (space charge). The broader the depletion region, the smaller the SCLC. The degree of SCLC may be predicted in association with the magnitude of the m_2_ value. From a simple calculation using Equation (7), since *V_bi_* increases from 1.15 eV for QLEDs-S1 to 1.22 eV for QLEDs-S2 and 1.6 eV for QLEDs-S3 in Figure 11, the relative ratio of the W_D_ will be W_D_(S1): W_D_(S2): W_D_(S3) = 1: 1.03: 1.17. This result well agrees with the decrease in the magnitude of m_2_ = 1.91 for QLEDs-S1, m_2_ = 1.56 for QLEDs-S2, and m_2_ = 1.54 for QLEDs-S3 as shown in Figure 17c. In the QLEDs-D1 and D2, the value of m_2_ is noticeably increased from m_2_ = 1.56 to m_2_ = 1.89 for the QLEDs-D1 and from m_2_ = 1.54 to m_2_ = 1.65 for the QLEDs-D2 respectively. As shown in Figure 15, two *V_bi_*’s are formed at the Ag/ZnO and ZnO/ZnO-OA interface. *V_bi_* at Ag/ZnO is the same value of 1.15 eV for both devices, but the second *V_bi_* is 0.07 eV for QLEDs-D1 and becomes larger as much as 0.45 eV for QLEDs-D2 respectively. The *V_bi_* = 0.07 eV at the interface of ZnO-OA (0.5 nm)/ZnO QDs is relatively smaller than *V_bi_* = 1.15 eV and thus will not significantly affect the SCLC. This assumption is strongly verified by the fact that m_2_ = 1.89 for the QLEDs-D1 shows an almost similar value to m_2_ = 1.91 for the OLED-S1. Compared to the QLEDs-D1, the *V_bi_* = 0.45 eV at the interface of ZnO-OA (0.5 nm)/ZnO QDs is not negligible and thus will have some influence on the SCLC by the formation of another depletion layer. Since the value of m_2_ = 1.65 for the QLEDs-D2 doesn’t noticeably increase from m_2_ = 1.54 for the QLEDs-S3, this conjecture can also be reasonably accepted. At higher bias voltage of about 2.5 V, the QLEDs-S1 shows m_3S_ = 3.71 until 5 V and m_3L_ = 13.8 until 7 V which are larger than m = 2. This means that the SCLC is converted into the TCLC. Overall, TCLC is governed by the density of interfacial traps and is also related to the magnitude of *V_bi_*. In the QLEDs-S1 having the *V_bi_* = 1.15 eV at the Schottky junction of Ag/ZnO, TCLC begins at around 2.5 V and shows m_3S_ = 3.71 until 5 V. This suggests that when the voltage surpasses a threshold voltage of about 2.5 V, the electrons will be captured by the induced traps distributed exponentially within the forbidden gap, therefore the electrical current increases rapidly. Another strong TCLC is observed at a voltage of 5 V and shows an abrupt increase of the slope up to m_3L_ = 13.8 until around 7 V. The sudden increase of the slope at about 5 V may be associated with the large change in the linear slope in the ln (J/V^2^)-(1/V) plot represented in Figure 17d. With increasing the voltage, the local electrical field-induced band bending gradually changes from square to triangular at the junction of Ag/ZnO, resulting in a much larger F-N tunneling current. The linear slope in ln (J/V^2^)-(1/V) of the QLEDs-S1 shows the value of *k* = 22 and rapidly increases at around 5 V, almost equivalent to the voltage of 5 V at which a strong TCLC occurs in Figure 17c. Contrary to the QLEDs-S1, negative values of the slope in the ln (J)-ln (V) plot for both the QLEDs-S2 and -S3 are observed in the voltage range of 2.5~7 V. The shape of the slope for the QLEDs-S2 is monotonously linear, but that for the QLEDs-S3 has an irregular zig-zag shape. Concerning the *V_bi_* = 1.22 and 1.6 eV for the QLEDs-S2 and -S3 not much different from *V_bi_* = 1.15 eV for the QLEDs-S1, TCLC should also occur similarly in the QLEDs-S2 and -S3, too. The current increase is suppressed in the negative slope region of 2.5~5 V, but it shows an abrupt current increase again after 7 V and m_3L_ = 15.3 and 17.2, which are similar to m_3L_ = 13.8 for the QLEDs-S1. This result is considered as if the strong TCLC that occurred in the QLEDs-S1 starts from 7 V. This voltage also agrees well with 7 V, where the linear slope increases abruptly in the ln (J/V^2^)-(1/V) plot as shown in Figure 17d. The origins of suppression of TCLC in the voltage of 2.5–5 V can be inferred from several reasons. The attached long chain of the OA ligand will play two roles. First, by reducing to some extent the number of electrons injected through the hopping mechanism into the inner ZnO core QDs from the Ag cathode. Second, by influencing the density of traps induced by surface defects. It is well known that ZnO QDs have commonly intrinsic defects such as oxygen vacancy (V_O_) or Zn interstitial (Zn_i_), which are mostly located at the surface of ZnO QDs. According to the previous studies [26,27,28], when ZnO QDs are adopted as ETL, the improved current induced by TCLC has been ascribed to the presence of V_O_. From the O1s core-level XPS spectra (Figure 5c), the relative chemical bond ratio of Zn-O: V_O_ = 49.7%: 20.5% in ZnO QDs, 9.9%: 20.6% in ZnO-OA (0.5 mL) QDs, and 2.9%: 20.6% in ZnO-OA (1.0 mL) QDs are estimated by Gaussian fitting, which confirms that V_O_ is included in all ZnO and ZnO-OA QDs. The ZnO-OA QDs are produced by forming a Zn acetate (Zn-O-C=O) bond in the chemical reaction between Zn interstitial in ZnO QDs and a carboxyl group in the OA ligand. As the amount of OA ligands increases, the number of Zn-OA (Zn-O-C=O) bonds also increases. This indicates that YL corresponding to the electron transition from Zn_i_ to the energy level (~0.9 eV from VB) of deep defects such as V_Zn_^2−^ or V_O_ becomes weak, but BL corresponding to the electron transition from ex-Zn_i_ to V_Zn_^−^ (~0.2 eV from VB) becomes relatively strong. V_O_ does not participate in the chemical reaction and will remain unchanged. Based on this assumption, TCLC induced by V_O_ should be observed in the voltage ranges of 2.5–5 V. Related to the above results, it can be expected that the negative slope of m_3N_ = −0.9 and m_3N_ = −0.97 in the low TCLC region results from the insufficient number of electron for filling the traps induced by V_O_. This lowered electron injection efficiency originates from the increase of the hopping mechanism by the long chain of OA ligands bound to the ZnO QDs surface. The irregular shape of the negative slope for the QLEDs-D1 suggests that the uniformity of OA ligands bound to the ZnO QDs surface is lower than that of the QLEDs-D2. However, a strong TCLC starts at 7 V, at which point the strong TCLC in the QLED-S1 disappears, and ends at 10 V. This phenomenon is supported because a steep increase of the linear slope in the ln (J/V^2^)-(1/V) plot for the QLEDs-D1 and-D2 (Figure 17c) happens at around 7 V and is caused by F-N tunneling. A negative slope in the ln(J/V^2^)-(1/V) plot for the QLEDs-D1 and -D2 is observed at the voltage region of 3–5 V, indicating that these are not enough high for electrons to fill the trap. As another reason, it cannot be ruled out that the increase of *V_bi_* from 1.15 eV for the QLEDs-S1 to 1.22 and 1.6 eV for the QLEDs-D1 and-D2 will also affect the variations of the TCLC. On the other hand, the QLEDs-D1 and -D2 having ETLs combined with ZnO QDs do not show any negative slope. The values of m_3S_ = 3.77 and m_3L_ = 13.3 for the QLEDs-D1 are very close to those of m_3S_ = 3.71 and m_3L_ = 13.8 for the QLEDs-S1. In the case of the QLEDs-D2, the value of m_3S_ becomes slightly small to 2.96, but that of m_3L_ increases to 14.8. The reappearance of the low TCLC in the QLEDs-D1 and -D2 can be explained by the lowered *V_bi_* of 0.07 eV and 0.45 eV and the increased number of electrons by combining ZnO QDs ETL. After the strong TCLC and F-N tunneling, the I-V data for the QLEDs-S1, -D1, and -D2 after 7 V can be well fitted with the slope of nearly m_1_ = 1. It seems all the traps are fully occupied, and the electrons can transport through the device without being trapped, leading to ohmic conduction after TCLC.

## 4. Conclusions

ZnO-OA QDs were synthesized by a chemical reaction of ZnO QDs solution precipitation at 70 °C with capping ligands of the OA. As the amount of OA increases from 0.5 mL to 1.5 mL, the crystalline structure, optical, and electrical properties of the ZnO-OA QDs were significantly modified. The increment of the attached OA ligands reduces the size of ZnO QDs and also deteriorates the crystallinity of ZnO QDs. The increase of chemical bonds of Zn-O-C=O formed between Zn interstitial of ZnO QDs and OA ligands induces the increase in the relative intensity of BL to YL of ZnO-OA QDs. Due to the electron transfer from ZnO to OA ligands, the work functions of 3.15 eV for ZnO QDs are reduced to 3.08 eV for ZnO-OA (0.5 mL) and 2.72 eV (1.0 mL), but the built-in potential (*V_bi_*) is reversely increased from 1.15 eV to 1.22 eV and 1.6 eV respectively. It is also noteworthy that the energy level of the VB for ZnO-OA QDs becomes deeper to 7.22–7.23 eV than 6.4 eV for ZnO QDs, which enhances the hole blocking efficiency. QLEDs with a single ZnO-OA QDs ETL show much lower EL than that of QLEDs with a single ZnO QDs ETL, but those with double ETLs of ZnO-OA/ZnO QDs show the much enhanced EL by 5~9% in luminance and 16~38% in current efficiency. The EOD using a double ETL of ZnO-OA (0.5 mL)/ZnO QDs shows the current density (J)-voltage (V) characteristic curve closer to the HOD. From the conduction mechanism, In the QLEDs-S2 and S3 adopting a single ZnO-OA QDs ETL compared to QLEDs-S1, the increase of current density in the voltage of 2.5–5 V through the TCLC related to V_O_ defect is largely restricted by a reduction in the electron injection due to increase of hopping conduction through OA ligands. However, the QLEDs-D1 and D2 using double ZnO-OA/ZnO ETLs show better EL performance by the combination of ZnO QDs having high conductivity and ZnO-OA QDs having high hole blocking efficiency.

## Figures and Tables

**Figure 1 nanomaterials-12-02038-f001:**
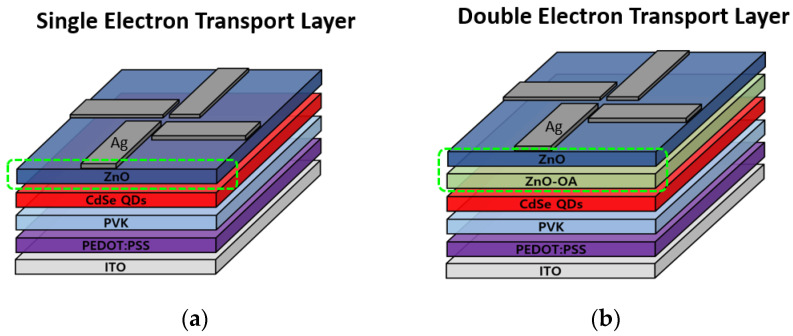
This Schematic cross-sectional view of (**a**) ITO/PEDOT:PSS/PVK//CdSe/ZnS//ZnO/Ag (single ETL) and (**b**) ITO/PEDOT:PSS/PVK//CdSe/ZnS//ZnO-OA/ZnO/Ag (double ETLs) QLEDs.

**Figure 2 nanomaterials-12-02038-f002:**
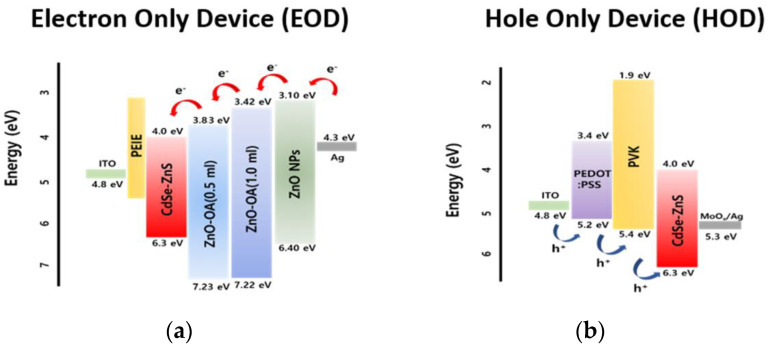
Electron energy level diagram of (**a**) EOD and (**b**) HOD.

**Figure 3 nanomaterials-12-02038-f003:**
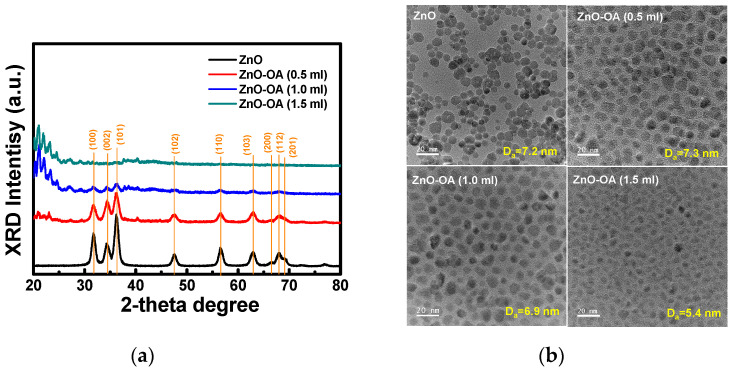
(**a**) XRD patterns and (**b**) HR-TEM images of ZnO QDs and ZnO-OA (0.5 mL, 1.0 mL, and 1.5 mL) QDs.

**Figure 4 nanomaterials-12-02038-f004:**
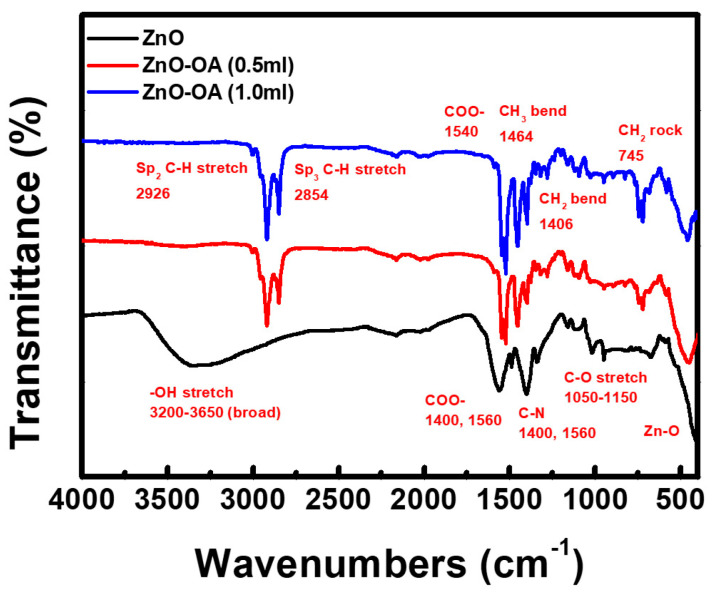
FT−IR spectra of ZnO QDs and ZnO−OA (0.5 mL, 1.0 mL, and 1.5 mL) QDs.

**Figure 5 nanomaterials-12-02038-f005:**
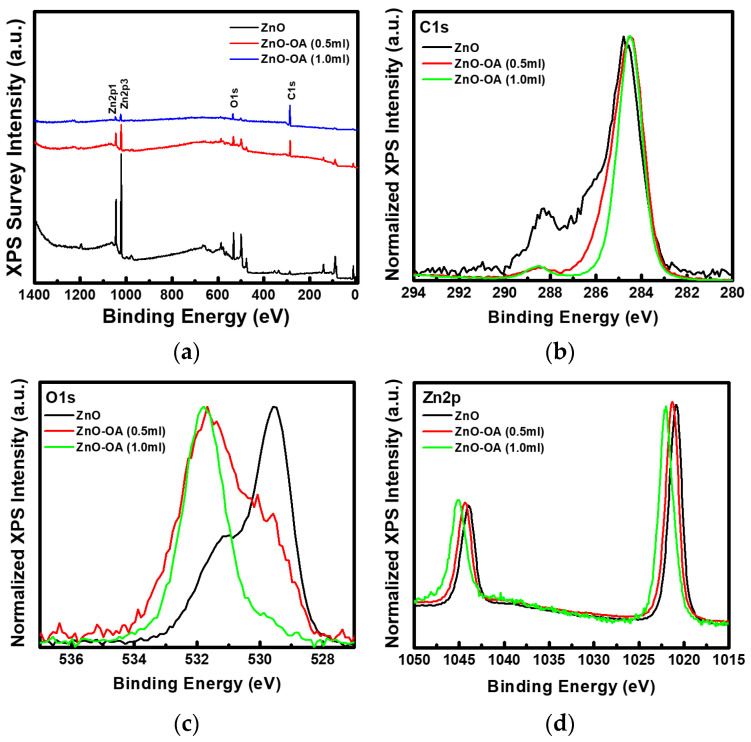
XPS (**a**) survey spectra, (**b**) C1s, (**c**) O1s, and (**d**) Zn2p core-level spectra of ZnO QDs and ZnO-OA (0.5 mL, 1.0 mL) QDs.

**Figure 6 nanomaterials-12-02038-f006:**
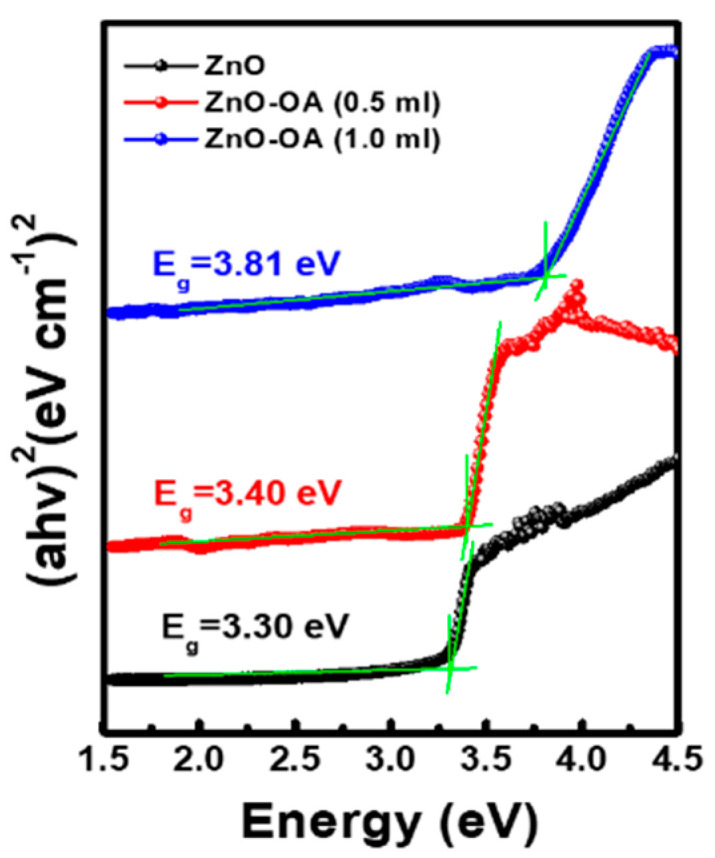
Tauc plot of ZnO QDs and ZnO−OA (0.5 mL, 1.0 mL) QDs.

**Figure 7 nanomaterials-12-02038-f007:**
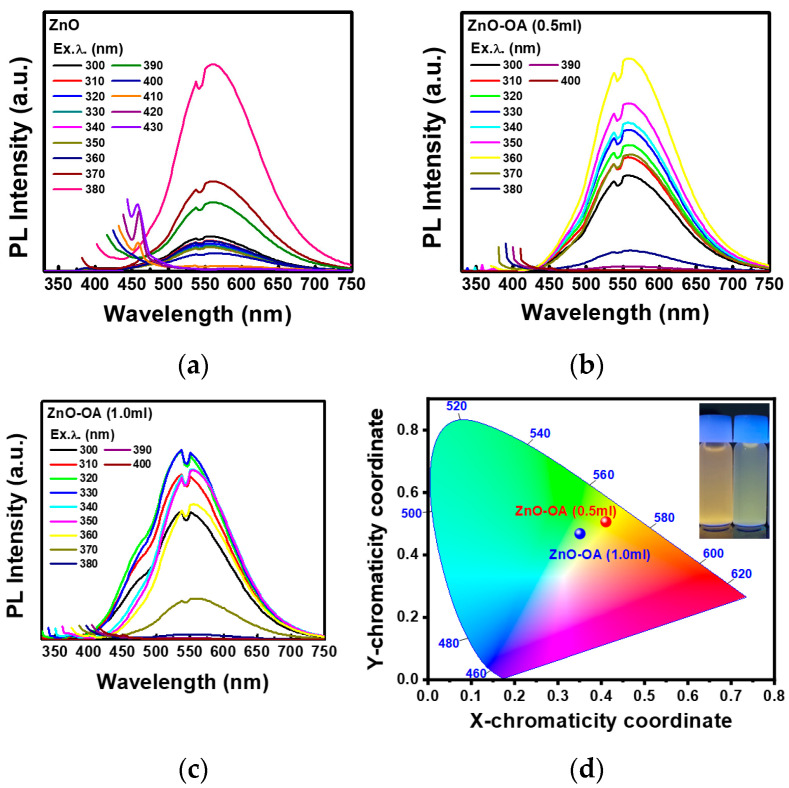
Spectral dependent Photoluminescence spectra of (**a**) ZnO QDS, (**b**) ZnO-OA (0.5 mL), and (**c**) ZnO-OA (1.0 mL) QDs. (**d**) CIE 1931 of ZnO and ZnO-OA (0.5 mL, 1 mL) QDs.

**Figure 8 nanomaterials-12-02038-f008:**
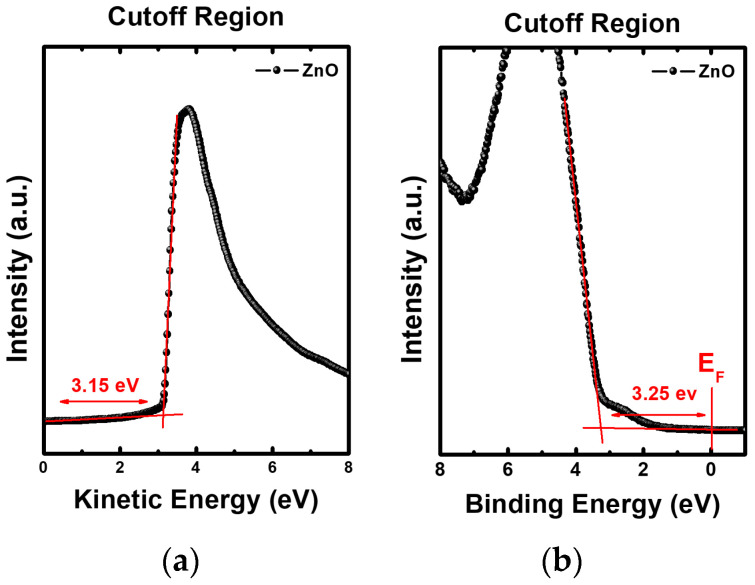
Secondary cutoff region of (**a**) ZnO QDs, (**c**) ZnO-OA (0.5 mL) QDs, and (**e**) ZnO-OA (1.0 mL), and valence region of (**b**) ZnO QDs, (**d**) ZnO-OA (0.5 mL) QDs, and (**f**) ZnO-OA (1.0 mL) QDs in UPS.

**Figure 9 nanomaterials-12-02038-f009:**
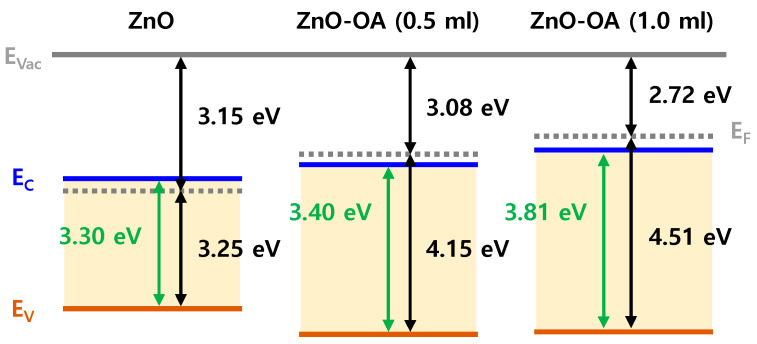
Electron energy level diagram of ZnO QDs, ZnO-OA (0.5 mL) QDs, and ZnO-OA (1.0 mL) QDs.

**Figure 10 nanomaterials-12-02038-f010:**
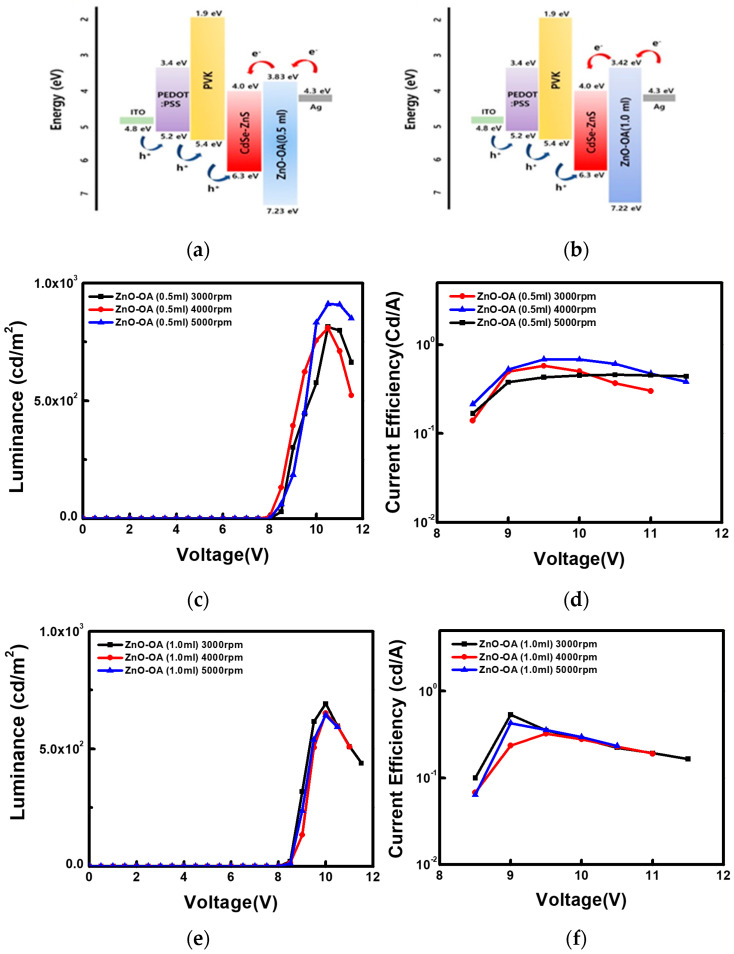
QLEDs structure with (**a**) ZnO−OA (0.5 mL) QDs and (**b**) with ZnO−OA (1.0 mL) QDs ETL. (**c**) Luminance−voltage curves and (**d**) current efficiency−voltage curves of QLEDs with different thicknesses of ZnO−OA (0.5 mL) QDs ETL. (**e**) Luminance−voltage curves and (**f**) current efficiency−voltage curves of QLEDs with different thicknesses of ZnO−OA (1.0 mL) QDs ETL.

**Figure 11 nanomaterials-12-02038-f011:**
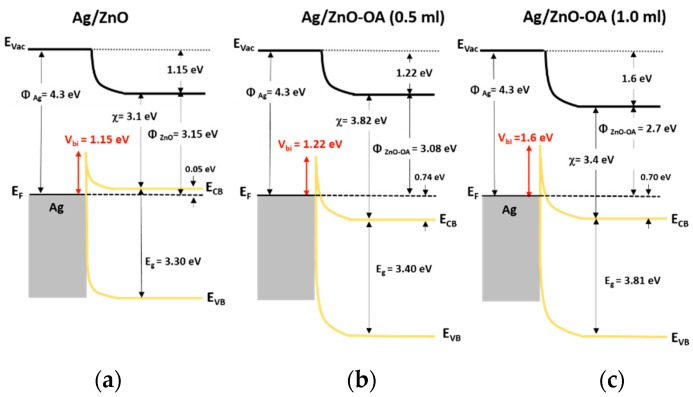
Induced built-in potential (*V_bi_*) at the interface of (**a**) Ag/ZnO QDs, (**b**) Ag/ZnO-OA (0.5 mL) QDs, and (**c**) Ag/ZnO-OA (1.0 mL) QDs.

**Figure 12 nanomaterials-12-02038-f012:**
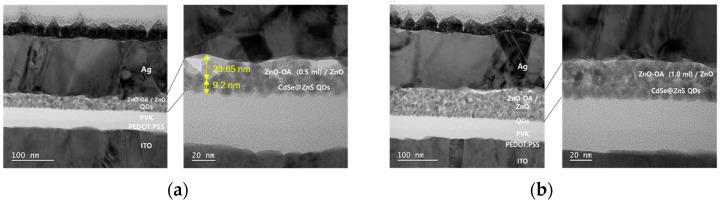
Cross-sectional HR-TEM images of QLEDs with (**a**) ZnO-OA(0.5 mL)/ZnO and (**b**) ZnO-OA (1.0 mL)/ZnO QDs double ETL. (**c**) EDS line mapping and line scan profile over the cross-sectional area of ITO//CdSe/ZnS//ZnO-OA (0.5 mL) QDs/ZnO QDs. AFM images over the surfaces of (**d**) ITO/CdSe-ZnS QDs, (**e**) ITO//CdS/ZnS//ZnO QDs, (**f**) ITO//CdSe/ZnS//ZnO-OA(0.5 mL) QDs/ZnO QDs, and (**g**) ITO//CdSe/ZnS//ZnO-OA(1.0 mL) QDs/ZnO QDs.

**Figure 13 nanomaterials-12-02038-f013:**
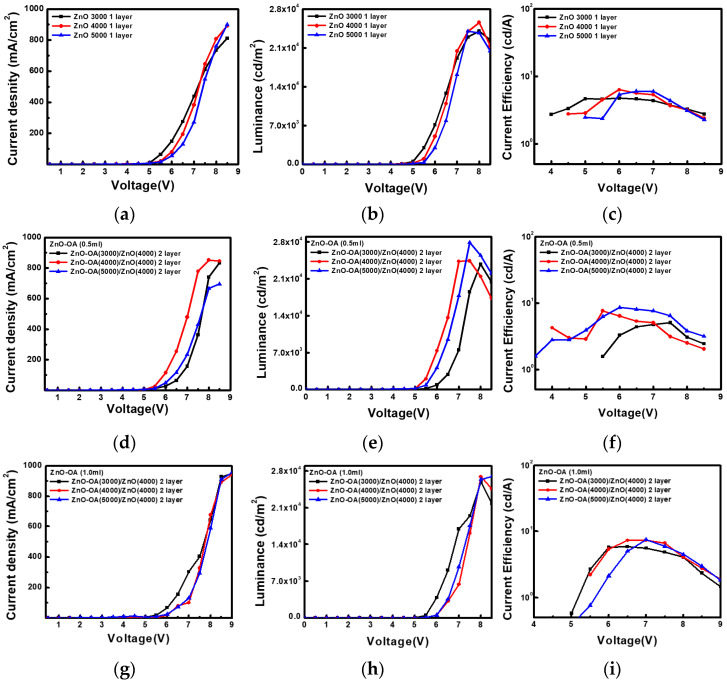
(**a**,**d**,**g**) current density (J)-voltage (V), (**b**,**e**,**h**) Luminance (L)-Voltage (V), and (**c**,**f**,**i**) Current efficiency-voltage (V) for QLEDs with a single ZnO QDs ETL and ZnO-OA (0.5 mL)/ZnO, and ZnO-OA (1.0 mL)/ZnO double ETLs respectively.

**Figure 14 nanomaterials-12-02038-f014:**
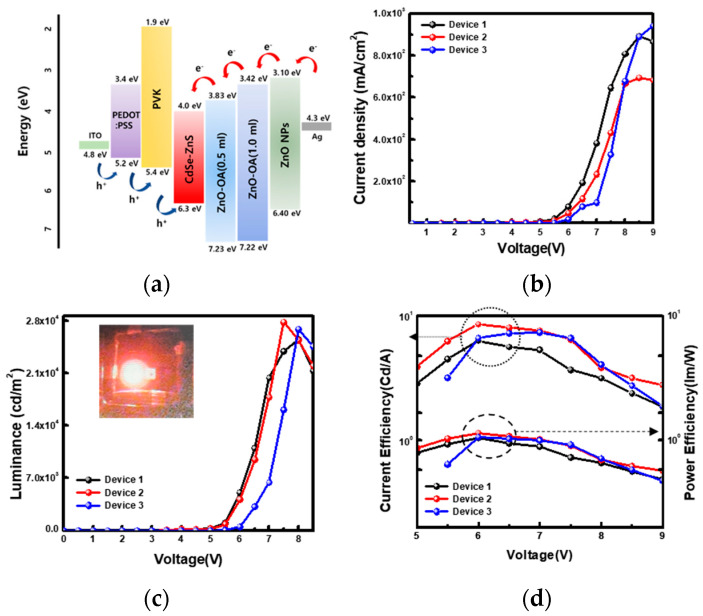
QLEDs structure with (**a**) ZnO-OA(0.5 mL, 1.0 mL)/ZnO-QDs ETLs. (**b**) current density (J)-voltage (V) for device 1 (QLEDs-S1), device 2 (QLEDs-D1), and device 3 (QLEDs-D2). (**c**) Luminance-voltage curves and (**d**) current efficiency (power)-voltage curves for device 1, 2 and 3.

**Figure 15 nanomaterials-12-02038-f015:**
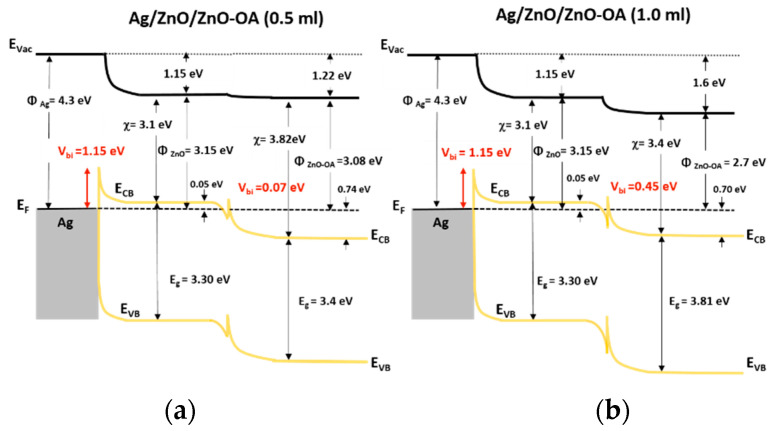
Induced built-in potential (*V_bi_*) at the interfaces of (**a**) Ag/ZnO-OA (0.5 mL)/ZnO QDs and (**b**) Ag/ZnO-OA (1.0 mL)/ZnO QDs.

**Figure 16 nanomaterials-12-02038-f016:**
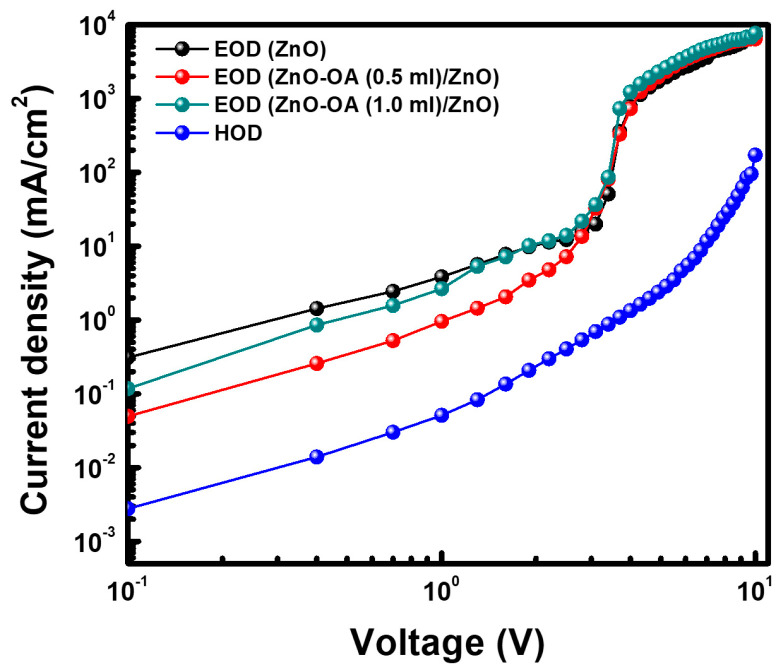
Current density (J)−voltage (V) curves of EODs and HOD.

**Figure 17 nanomaterials-12-02038-f017:**
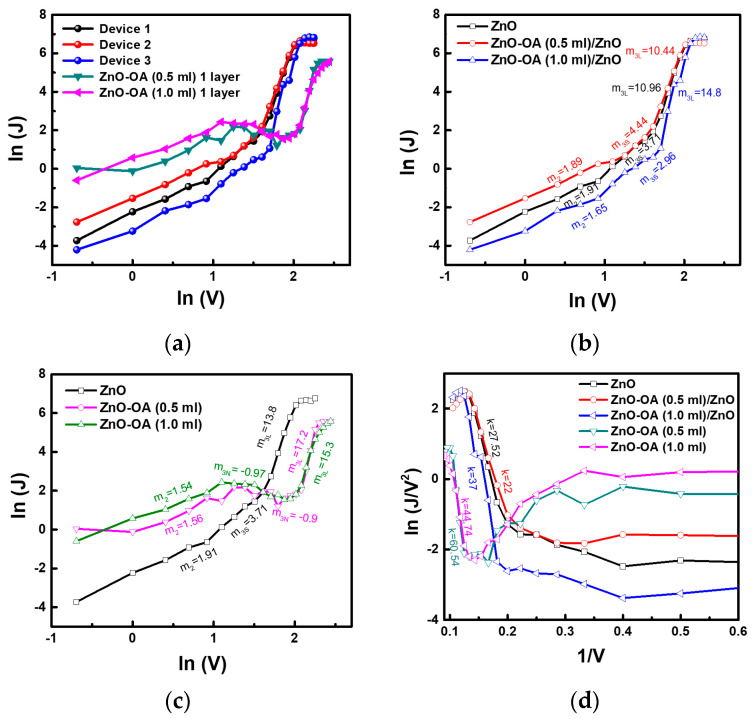
(**a**) current density (J)−voltage (V) curves for QLEDs−S1, −S2, −S3, −D1 and −D2. ln(J)−ln(V) curves of (**b**) QLEDs−S1, −D1, and −D2, and (**c**) QLEDs−S1, −S2, −S3. (**d**) ln(J/V^2^)(1/V) for QLEDs−S1, −S2, −S3, −D1, and −D2.

**Table 1 nanomaterials-12-02038-t001:** QLEDs performance of a single ETL and double ETLs structure at various thicknesses.

Layer	Spin Coating (rpm)	Turn-On (V)	Maximum Luminance (cd/m^2^)	Current Efficiency (cd/A)
ZnO-OA (0.5 mL)/ZnO	3000/4000	3.5	23,760	5.08
ZnO-OA (0.5 mL)/ZnO	4000/4000	3.0	24,430	7.64
ZnO-OA (0.5 mL)/ZnO	5000/4000	3.5	27,890	8.57
ZnO-OA (1.0 mL)/ZnO	3000/4000	4.0	25,940	5.84
ZnO-OA (1.0 mL)/ZnO	4000/4000	4.5	26,923	7.37
ZnO-OA (1.0 mL)/ZnO	5000/4000	5.0	26,980	7.48
ZnO single layer	4000	4.5	25,570	6.32

## Data Availability

The authors confirm that the data supporting the findings of this study are available within the article and its Appendix A.

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
