# Peer review of "Enhanced Luminance of CdSe/ZnS Quantum Dots Light-Emitting Diodes Using ZnO-Oleic Acid/ZnO Quantum Dots Double Electron Transport Layer"

_nanomaterials, 2022, doi:10.3390/nano12122038_

Round 1

Reviewer 1 Report

Authors provided an extensive research study, with a great amount of experimental work on synthesis of ZnO QDs. Authors studied different aspects of prepared structures, including optical and electrical properties. The study is highly multi-disciplinary.

The manuscript is well prepared with a care. From time-to-time is was difficult to go through all the details that were provided by Authors, but the overall readability is high. The methodology employed in this work was adequate with a clear strcuture and description. Results presentation and discussion was solid and explain in detail.

Although the study described in this manuscript is deep and comprehensive, I am not sure about the overall work imapct on the community and potential readers. I assume, this type of investigations can find readers in a very narrow field (I assume mostly for experts, and not for wider audience).

Some suggestions/questions to this work are provided here:

- The quality of provided figures should be improved. They should be at least with 300 dpi. Some of the figures are also difficult to read, when the paper is printed.

- The manuscript would be strengthen by a benchmark table comapring results/approaches from the other groups with those presented here.

- Can Authors comment on the critical aspects or limitations with respect to the crystalline quality? 

- Short paragraph with respect to the intended applications would be more than helpful.

Author Response

Dear,

Please find the attached response.

Reviewer 2 Report

In this manuscript, the luminance of a QLEDs was enhanced by 5~9 % with an additional electron transport layer ZnO-OA, which decreases the work function for ZnO by charge transfer from ZnO to OA ligands. Following are some comments to make this manuscript more clear for better understanding.

(1) ZnO is dissolved in ethanol while ZnO-OA is dissolved in 2-propanol, could the author comment on how to fabricate the separate ZnO-OA and ZnO layers?

(2) Figure 3 (b) is missing in PDF file.

(3) please double check Figure 10. Figure 10 e and f should be ZnO-OA (1.5 ml) instead of ZnO-OA (0.5 ml) according to the figure caption. However, it was 1.0ml in the main text. It is very confusing. Additionally, Figure c/d and Figure e/f looks the same to each other now.  

(4) Line 298 on page 9, there is no Figure g while authors list “As shown in Figure 10 d,g”

Author Response

(The authors gave the same response as above.)

Round 2

Reviewer 2 Report

Authors have responded to all the comments and made the appropriate corrections. This paper is now ready for publication.